# Targeted NGS-Based Analysis of *Pneumocystis jirovecii* Reveals Novel Genotypes

**DOI:** 10.3390/jof8080863

**Published:** 2022-08-17

**Authors:** Dora Pungan, Taylor Eddens, Kejing Song, Meredith A. Lakey, Nicolle S. Crovetto, Simran K. Arora, Shahid Husain, Jay K. Kolls

**Affiliations:** 1Center for Translational Research in Infection and Inflammation, John W. Deming Department of Medicine, Tulane University School of Medicine, New Orleans, LA 70112, USA; 2Children’s Hospital of Pittsburgh, Pittsburgh, PA 15201, USA; 3Biospecimen and Core Research Laboratory, Department of Research, Ochsner Health System, New Orleans, LA 70121, USA; 4Multi-Organ Transplant Program, Division of Infectious Diseases, Department of Medicine, University Health Network/University of Toronto, Toronto, ON M5G 2N2, Canada

**Keywords:** *Pneumocystis*, genotyping, pneumonia

## Abstract

*Pneumocystis jirovecii* is an important etiological agent of pneumonia that is underdiagnosed due to the inability to culture the organism. The 2019 PERCH study identified *Pneumocystis* as the top fungal cause of pneumonia in HIV-negative children using a PCR cutoff of 10^4^ copies of *Pneumocystis* per mL of sample in nasopharyngeal/oropharyngeal (NP/OP) specimens. Given that *Pneumocystis* consists of an environmental ascus form and a trophic from (the latter is the form that attaches to the lung epithelium), it is possible that life-form-specific molecular assays may be useful for diagnosis. However, to accomplish this goal, these assays require genotypic information, as the current fungal genomic data are largely from the US and Europe. To genotype *Pneumocystis* across the globe, we developed an NGS-based genotyping assay focused on genes expressed in asci as well as trophs using PERCH throat swabs from Africa, Bangladesh, and Thailand, as well as North American samples. The NGS panel reliably detected 21 fungal targets in these samples and revealed unique genotypes in genes expressed in trophs, including Meu10, an ascospore assembly gene; two in mitochondrial gene ATP8, and the intergenic region between COX1 and ATP8. This assay can be used for enhanced *Pneumocystis* epidemiology to study outbreaks but also permits more accurate RT-CPR- or CRISPR-based assays to be performed to improve the non-bronchoscopic diagnosis of this under-reported fungal pathogen.

## 1. Introduction

*Pneumocystis jirovecii* is the leading fungal cause of pneumonia in young children regardless of HIV status [1]. Symptomatic clinical infection, however, is associated with immunosuppression due to primary immune deficiency or drugs/treatment regimens that suppress the immune system [2,3]. Due to the inability to culture *Pneumocystis* sp., as well as a lack of non-invasive diagnostics, the epidemiology of this infection likely underestimates the true incidence and prevalence of disease. Autopsy data showed that *Pneumocystis* is quite commonly detectable in lung tissue in children under the age of 2 [4]. The PERCH study showed that *Pneumocystis* was a major cause of pneumonia in HIV-negative children, particularly in children under the age of 2 [1]. Importantly, in this study, the authors used a threshold of 10^4^ copy number of *Pneumocystis* per mL of sample as the threshold for diagnosis using throat/NP swabs. These data suggested that non-bronchoscopic diagnostics could be developed using this type of sample collection.

A limitation of mitochondrial large subunit of ribosomal RNA (mtLSU) testing is that this target may not discriminate between colonization and infection. This is one reason why the threshold of 10^4^ copies was chosen as a disease classifier in PERCH. Moreover, other targets were used for Pneumocystis epidemiological studies [5]. Current genotyping studies focused on multilocus sequence typing performed on DNA [5,6] targeting superoxide dismutase, mtLSU, or cytochrome b.

In an effort to assess life-form-specific transcripts, we recently annotated the *P. murina* genome and conducted transcriptomic analyses of the ascus as well as the troph with flow-cytometry-sorted organisms [7]. These RNA data showed that the ascus and troph differed in their RNA transcriptomes. We found specific genes that were preferentially expressed in the troph, such as serine protease and many mitochondrial electron genes (ATP synthase 1 and 8, and cytochrome c oxidase subunit 1 and 3) [2,7], whereas genes such as glucan synthetase 1 (GSC1) and Arp9 were more highly expressed in the ascus. We validated SP, Arp9, and GSC1 in vivo using ascus-specific depletion with echinocandin treatment [7]. Thus, a troph-specific RNA target may be diagnostic of infection whereas RNA targets expressed in the ascus may be more consistent with colonization. It is thought that the ascus is more frequent in colonization [8,9] and that the troph is more frequent in active infection, given that the latter attaches to the alveolar epithelium [2,10]. Given that the current genomic data are largely from the US and Europe, in this study, we sought to genotype a series of genomic and mitochondrial targets in the DNA (Table 1) that would be potentially useful in designing geographic-specific reverse transcription-polymerase chain reaction (RT-PCR)- or clustered regularly interspaced short palindromic repeats (CRISPR)-based RNA detection assays in throat swabs. The rationale of the present study was to determine polymorphisms in these molecular targets, as these could affect the design and fidelity of these types of diagnostic efforts. To accomplish this, we tested this next-generation sequencing (NGS)-based genotyping assay on samples from PERCH as well as clinical samples collected in North America.

## 2. Materials and Methods

### 2.1. Human Samples

For this cross-sectional study, we used 23 individual *Pneumocystis*-positive samples from the PERCH study, specifically, samples from Bangladesh, Mali, Zambia, and Thailand. North American samples (n = 13) came from excess bronchoalveolar lavage fluid from clinical bronchoscopies from University Health Network (Toronto, ON, Canada) or Ochsner Medical Center (New Orleans, LA, USA). A demographic table of the samples is shown in the Appendix A.

### 2.2. DNA Extraction

For the nasopharyngeal and oropharyngeal (NP-OP) samples from the PERCH study, the total nucleic acids from each sample were provided, and the procedure for the preparation is included in their Laboratory SOP [1]. The North American samples were received as raw bronchoalveolar lavage fluid, and total nucleic acids were extracted using PrepSEQ Nucleic Acid Extraction Kit (Cat. Nos. 4480466 and 4428176; Thermo Scientific, Applied Bio-systems, Waltham, MA, USA) following the manufacturer’s protocol. The concentration of DNA in all the samples was determined with a Qubit Fluorometer using a Qubit dsDNA High Sensitivity assay kits (Cat. Nos. Q32851 and Q32854) following the manufacturer’s protocol.

### 2.3. Library Preparation

Next-generation-sequencing libraries from the DNA samples were prepared using a custom rhAmpSeq targeted sequencing panel using reagents from the rhAmpSeq Library kit (Cat. No. 10000064; Integrated DNA Technologies) following the manufacturer’s protocol. The rhAmpSeq workflow includes two PCR amplifications to generate the libraries. The first PCR includes RNase H-dependent PCR (rhPCR) primers along with the RNase H2 enzyme to perform rhPCR, where rhPCR is a nucleic-acid amplification method that provides increased target specificity over traditional PCR. As opposed to traditional DNA primers used in PCR and qPCR, rhPrimers contain a single RNA base (rN) and a 3′ blocking group. This latter moiety can only be cleaved after annealing to the matched target sequence. This reduces the risk of non-target amplification, such as primer dimers. The amplification was performed using Taq polymerase, as it is not recommended to use High-Fidelity polymerase, since proofreading enzymes with 3′–5′ exonuclease activity can remove the 3′ blocking group, which could lead to non-specific amplification. The second PCR in the rhAmpSeq workflow is indexing PCR, which amplifies rhAmp PCR amplicons using i7 and i5 index primers to complete the library building process. The rhAmpSeq system includes purification after both PCR steps using Agencourt Ampure XP Beads (Beckman Coulter, Pasadena, CA, USA; A63880). Targets were chosen based on prior transcriptomic data [7], including genes enriched in trophs including serine protease (T551_01755), Meu10 (T551_02533), and T551_03038; asci including ARP9 (T551_03199) and GSC1 (T551_02309); and several mitochondrial genes (the full list of targets and primers used for library generation is in Table 1). The total DNA input used was 10 ng. If the sample’s concentration was less than 10 ng/uL, a total of 11 uL of the sample was added, and no IDTE buffer was used for dilution. 

### 2.4. DNA Sequencing 

Final libraries were quantitated using a Qubit dsDNA HS assay kit (Thermo Fisher Scientific; Pittsburgh, PA Q32854, USA). The quality of the libraries was determined by running each on Agilent 4150 TapeStation using a DNA 1000 kit (Agilent, Santa Clara, CA, USA; #5067-1504). Smear analyses were performed using Agilent TapeStation Software (Version 3.1) with a range of 150–700 bp to determine the average size of each library. Size and concentration were then used to calculate the molarity of each library. All libraries were pooled and denatured following the standard normalization method in the Illumina Denature and Dilute Libraries Guide for NextSeq 550 System (Illumina #15048776, San Diego, CA, USA) and MiniSeq System (Illumina #1000000002697). Finally, denatured libraries were loaded onto an Illumina NextSeq 550 v2 Mid-Output reagent cartridge (#20024907) at a final concentration of 0.8 pM in HT1 buffer or Illumina MiniSeq High-Output reagent cartridge (Illumina #20024907) at a final concentration of 1.44 pM in HT1 buffer. Denatured PhiX control library v3 (#FC-110-3001) was also included at a 1% concentration. Single-end 148 bp dual-index sequencing was performed on an Illumina NextSeq Mid 150 flow cell v2.5 (#20024904) or Illumina MiniSeq High-Output 150 flow cells (#FC-420-1002), yielding approximately 50 M or 2 M reads per sample, respectively.

### 2.5. Sequencing Analysis

Using SeqMan NGen 17 Genome Assembly Software (part of Lasergene 17 software, DNASTAR, Inc., Madison, WI, USA), new assemblies were created from the fastq files generated from each sample. The workflow for the assembly to generate the SNP table was variant analysis/resequencing and NGS-based and then amplicon, gene panel, and exome. The reference genome used for read mapping was the whole genome of *Pneumocystis jirovecii* and the *P. jirovecii* genome assembly strain RU7 (NCBI accession number GCF_001477535.1). The reference genome used for the *Pneumocystis jirovecii* mitochondrial genome was the *Pneumocystis jirovecii* mitochondrion (complete genome from NCBI; accession number NC_020331.1). When inputting sequences, the fastq files were added, and the experiment setup was multi-sample; the samples were run in single assembly. The analysis options used were: haploid genome; variant detection mode; high SNP filter stringency, which includes 75% as the minimum variant percentage; and “P not ref” of 99.9%, which is the probability that the base does not match the reference. Once the assembly project was run, ArrayStar Software v3.0 was used to open the ArrayStar project file that was generated from the assembly pipeline. For SNPs identified in the coding sequence, we conducted potential functional analyses with the PredictSNP tool (https://loschmidt.chemi.muni.cz/predictsnp/; accessed on 27 May 2022) [11]. 

The assembly files generated from the above-explained pipeline were used in SeqMan Ultra 17 software, which provided the average-depth calculation across features. Using the View menu and selecting features and then the contig of interest, the resulting table provided the average coverage depth for the corresponding genes. 

## 3. Results

### 3.1. Library Prep Validation

We developed this IDT library to initially genotype both troph and ascus targets that may be useful for assaying the relative proportion of these two life forms in clinical samples (Table 1). To this end, we tested the library prep on 23 samples from PERCH [1] and 13 samples from North America. The sample demographics are depicted in Appendix A. The MiniSeq output ranged from as few as 4170 reads to a maximum of 14,836,268 reads per sample Appendix A. *P. jirovecii* mapped reads varied from 17 mapped reads to 3,122,534 (including reads that mapped to the genome as well as mitochondrial targets). The median of the percentage of reads mapping to *P. jirovecii* was 60.96% Appendix A. All the targets amplified in nine or more of the samples. A total percentage of 86.11% of the samples had at least 50% of the targets detected. The median percentage of targets detected was 92.86%. The average ratio of the percentage of mapped *P. jirovecii* mitochondrial reads to mapped *P. jirovecii* whole-genome reads was 84.63 to 15.37.

### 3.2. Meu 10 Genotype

With this NGS panel, we identified two SNPs in T551_02533: C > T at position 194,152 and a non-synonymous SNP, G > A at position 194184 (Figure 1). This latter SNP predicts an amino-acid change at position 232 with S > N variant. Interestingly, these two SNPs had strong concordance (100% concordance) in samples from Bangladesh, Mali, and Zambia. This genotype was present in 1/6 samples from Bangladesh, 3/6 samples from Mali, and 2/6 samples from Zambia but was absent in samples from Thailand or North America. Meu10 is a GPI-anchored protein that has been implicated in ascospore assembly in other fungi [12]. The analysis with PredictSNP [11], which is a bioinformatic tool to estimate deleterious effects on the coded protein, predicted largely a neutral effect of protein function, but PolyPhen1/2 predicted a 54–59% deleterious effect. Within the PERCH cohort, this genotype was not associated with the age of infection.

### 3.3. FourF5 Genotype

We identified several SNPs in putative Four F5 protein or T551_00896 Appendix A. This included a G > A substitution at position 577947 and a C > T substitution at position 578,000. These variants were observed in samples from Bangladesh, Mali, Thailand, Zambia, and North America Appendix A. There was an 89% concordance rate of these two SNPs being present within the same samples, suggestive of possible genetic linkage. A complete list of SNPs identified is in Appendix A. 

### 3.4. Mitochondrial Genotypes 

We also identified a unique set of SNPs in COX1 and ATP8 that showed a very high level of concordance in multiple samples (Figure 2). These variants consisted of C > A and C > T at positions 17847 and 17850, which are in the intergenic regions between COX1 and ATP8, and G > T at position 17928 in ATP8. These three variants were present in 5/6 samples from Zambia, 4/13 from North American, but only 1/6 from Mali and Thailand. This latter SNP results in a phenylalanine replacement of valine at codon 21 of the protein. The analysis with PredictSNP [11] predicted largely a neutral effect of protein function, but PhD-SNP predicted a 61% likelihood of a deleterious effect. Within the PERCH cohort, this genotype was not associated with the age at infection. For the intergenic SNPs, there was a high concordance of these two SNPs within the same sample, with 10 of 13 carrying both SNPs, respectively (Figure 2). Of the 13 samples carrying the 17850 C > T SNP, 11 of the samples also had the G > T SNP at position 17928 in ATP8 (Figure 2). The 17850 C > T SNP was present in at least one sample from every region but was most prevalent in samples from Zambia (Figure 2). We also identified SNPs in the intergenic regions between nad4l and nad5. These variants as well as the full SNP table for mitochondrial SNPs are listed in Appendix A.

## 4. Discussion

The release of *Pneumocystis* genomes over the last decade has greatly facilitated transcriptomic and genotyping approaches to understand organism heterogeneity and pathogenesis [7,8,13,14,15]. Alanio et al. recently published on six-target short tandem repeat (STR) markers, and with this approach, the authors noted the geographic clustering of genotypes in Europe. Moreover, there was a relationship between the genotype and underlying disease, as genotypes differed in HIV versus renal-transplant subjects [15]. Although we saw some evidence of geographic clustering with the mitochondrial SNPs, this would need to be replicated with a larger sample size. 

Charpentier and colleagues recently published on an NGS-based multilocus sequence typing (MLST) study in the context of a PCP outbreak [16] at a single clinical site. In this study, the authors targeted three genes, the superoxide dismutase (SOD), mtLSU, and cytochrome b (CYTB) genes. These authors designed an assay that had average amplicon sizes of ~750 bp. Notably, this study was performed on adult subjects with relatively high fungal burdens ranging from 10^4^ to 10^9^ copies per mL (based on the PCR of the major surface glycoprotein gene). 

Our study focused on the pediatric PERCH cohort as well as adult samples from North America. We developed an NGS assay to cover more diverse targets based on prior transcriptomic work to target genes that are expressed in asci, such as ARP9 and GSC1, as well as genes that are enriched in trophs [7]. However, we chose smaller amplicons and designed primers based on homology between these targets in *P. jirovecii*, *P. carinii*, and *P. murina* (Table 1). Despite these smaller amplicons, we identified several genotypes that may be useful in tracking the evolution of the organism as well as testing outbreaks. Moreover, some of these SNPs are coding variants. Due to the inability to culture *Pneumocystis,* the functional consequences of these variants may need to be tested in other fungi (Meu10 or T551_02533). The S-to-N variant was also found in *Pneumocystis* sp. *‘macacae’* (accession: KAG5519132.1), whereas *Pneumocystis murina* has an S-to-T variant in this position. This S-to-T variant is also present in *Schizosaccharomyces octosporus* (accession: XP_013019364.1) and *S. pombe* (accession NP_001342757.1), where it is required for spore-wall maturation [12]. Interestingly, β-1,3 glucan is abnormally localized in Meu10-deficient *S. pombe,* making affected specimens incapable of sporulation. Thus, this yeast strain may be suitable for future complementation studies. 

The phenylalanine substitution for valine in ATP8 was reported in NCBI sequences for *P. jirovecii* (accession: CCJ32492.1) as well as *Pneumocystis* sp. *‘macacae’* (accession: QTK22320.1). The homology of this protein in a culturable yeast such as *Saccharomyces* spp. is weak; thus, complementation studies for this gene may be difficult.

Taken together, we report a new NGS genotyping platform that can be run on BAL or throat swab samples that can be run on smaller instruments, such as MiniSeq. We think that these data are important for the mycology community, and we hope this aids the field in determining the origins of *P. jirovecii* as well as its geographic heterogeneity. 

## Figures and Tables

**Figure 1 jof-08-00863-f001:**
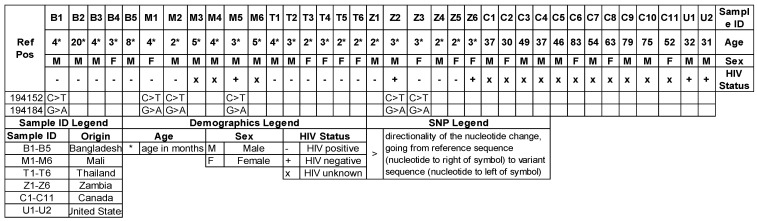
Meu10 SNPs by country/sample. All SNPs have Ref ID NW_017264785.1; Transcript ID XM_018374796.1; Gene ID T551_02533; *P. murina* ortholog Meu10.

**Figure 2 jof-08-00863-f002:**
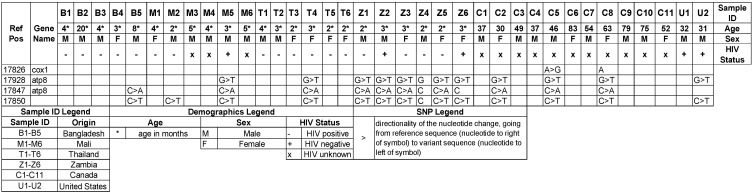
Mitochondrial SNPs identified by country/sample. The SNPs at reference position 17850 are within the intergenic regions between cox1 and atp8. All SNPs have Ref ID NC_020331.1.

**Table 1 jof-08-00863-t001:** NGS targets and primers used for NGS library generation.

Sequence Name	Target: Gene Name/Murina Ortholog	Target Length	Sequence 5′ to 3′	Primer Notes
RHH.W4300B6A5BDF041	N/A	N/A	N/A	Custom rhAmpSeq™ FWD Panel
RH.CF78E0180F584D9Z0Z.F	T551_03199/Arp9	134	/rhSeq-f/TCA TAG CCC ACA TGA ATA ACA ATT rCCA GA/GT4/	Forward
RH.A2C31F6013C24B7Z0Z.F	T551_03199/Arp9	148–152	/rhSeq-f/GTT CAA TGC TGA CAC TGT TTA TCrG CCA T/GT4/	Forward
RH.1B63D59F4F974A1Z0Z.F	T551_03038/meiotically up-regulated gene 117 protein	138–142	/rhSeq-f/GGT TTC TGA GAA AGA TTT AGC CAT ATrG TAA C/GT1/	Forward
RH.FD38EF6B8AEA421Z0Z.F	T551_00303/GTP binding	148–150	/rhSeq-f/GAT CCA CTG TCC AGA CAA TTrG CAT C/GT1/	Forward
RH.103374AC8028435Z0Z.F	T551_02533/Meu10	148	/rhSeq-f/ATA GGA GTG GTT TCC GTC TTT rCTT TC/GT3/	Forward
RH.E0CE477C4B19464Z0Z.F	T551_00896/Four F5 protein	150–153	/rhSeq-f/GAG AAA ATG ATA GAC TTA GGG CTrC AAA A/GT4/	Forward
RH.316DF1391E794B9Z0Z.F	T551_01756/Acrl	148–149	/rhSeq-f/CAG AAG ATG AAT TGC CAC CAA rACA TT/GT4/	Forward
RH.E45C51A8E1AB462Z0Z.F	T551_01755/Serine protease	110–112	/rhSeq-f/CAT CTG CAG GTA AAG ACA AGA AArC CAG A/GT1/	Forward
RH.AD8B9A3AFA344A3Z0Z.F	T551_01830/heat shock protein 70	142–144	/rhSeq-f/CCC GAA TTT CCG TCC AAT rCAG AC/GT2/	Forward
RH.47196AD4997D471Z0Z.F	T551_02650/SUMO-targeted ubiquitin-protein ligase E3 Slx8	136–138	/rhSeq-f/TGA CAC ACA AAG GCC ATT TAT TrCT GTC/GT1/	Forward
RH.AEC03A6FD9A64E0Z0Z.F	T551_03158/DNAJ	135–136	/rhSeq-f/CCG AAT TGG AAC TCA CTT TAC TrCG AAG/GT3/	Forward
RH.E8A3CF8496644ACZ0Z.F	T551_02309/GSC-1	129–134	/rhSeq-f/TCT GGA GTA AAT CGA ACT TGA TTT rGCT TC/GT4/	Forward
RH.8DF9507B0841432Z0Z.F	mito NAD6	149–153	/rhSeq-f/CAA TTC TGG TTG TAT CTT CAA GAA ATC rCTG TA/GT2/	Forward
RH.3E370038B0CB4B3Z0Z.F	mito NAD1	151–156	/rhSeq-f/GCT TTA TTA GGT TCC TTG TGA AGT ArCT GCT/GT3/	Forward
RH.DDFCD6AD33494FBZ0Z.F	cytochrome c oxidase subunit 1/ atp8	155–160	/rhSeq-f/TGA TTT CAC CAC CAG CTT TCrC ATG C/GT1/	Forward
RH.FF72BC52B1E14BCZ0Z.F	atp6	155–160	/rhSeq-f/CTT CAC ATC TGG TCT TTA CTG TCrG CTT T/GT3/	Forward
RH.E67274DA8AD04D9Z0Z.F	cytochrome c oxidase subunit 3	151–156	/rhSeq-f/CTA GCT TTT ACT CTT TTA CAG GGT rGTG GA/GT4/	Forward
RH.EF9CF32F289649AZ0Z.F	atp9	149–152	/rhSeq-f/GGT TCA GGG TTA GCT ACA ATT rGGA TT/GT4/	Forward
RH.99229D6013E14CBZ0Z.F	NADH dehydrogenase subunit 4L/nad5	151–152	/rhSeq-f/TCC ATG GCT TTA GAT GAT TTA GAA GrGA CAG/GT1/	Forward
RH.795964E1E42344EZ0Z.F	Nad4	125–128	/rhSeq-f/TCA CTC CTT TGG TCT ATA CTG TrCT GTG/GT1/	Forward
RHH.W4300B6A5BDF041	N/A	N/A	N/A	Custom rhAmpSeq™ REV Panel
RH.CF78E0180F584D9Z0Z.R	T551_03199/Arp9	134	/rhSeq-r/CGT GAT AAA GAA CTT GCA ACA TGrC ACA T/GT1/	Reverse
RH.A2C31F6013C24B7Z0Z.R	T551_03199/Arp9	148–152	/rhSeq-r/GAG AGT ATT TGG CTC AAA AAG AAG rCGG AG/GT4/	Reverse
RH.1B63D59F4F974A1Z0Z.R	T551_03038/meiotically up-regulated gene 117 protein	138–142	/rhSeq-r/TCT ACT CTA TTA ATG TAT TCT GAC CCA rCAA GG/GT4/	Reverse
RH.FD38EF6B8AEA421Z0Z.R	T551_00303/GTP binding	148–150	/rhSeq-r/GGT TAT GAC AAT TTC GCC GArC CGT T/GT2/	Reverse
RH.103374AC8028435Z0Z.R	T551_02533/Meu10	148	/rhSeq-r/TGA CCA AGT CGC TGT TAT TGT rATA TG/GT2/	Reverse
RH.E0CE477C4B19464Z0Z.R	T551_00896/Four F5 protein	150–153	/rhSeq-r/GAT GTT TGC TTG GCT AAG GTA rCAG AT/GT1/	Reverse
RH.316DF1391E794B9Z0Z.R	T551_01756/Acrl	148–149	/rhSeq-r/TCC CAT CGT AAA AGG CGT AAT rCTT GA/GT1/	Reverse
RH.E45C51A8E1AB462Z0Z.R	T551_01755/Serine protease	110–112	/rhSeq-r/TGA ACA AAT GAA TGC GAC AAT AGT rCCC AA/GT2/	Reverse
RH.AD8B9A3AFA344A3Z0Z.R	T551_01830/heat shock protein 70	142–144	/rhSeq-r/TCA AGG GAA TCG AAC CAC ArCC ATC/GT4/	Reverse
RH.47196AD4997D471Z0Z.R	T551_02650/SUMO-targeted ubiquitin-protein ligase E3 Slx8	136–138	/rhSeq-r/TTT GTT TGG TGC AAG CAT TAT TTrC AAG G/GT2/	Reverse
RH.AEC03A6FD9A64E0Z0Z.R	T551_03158/DNAJ	135–136	/rhSeq-r/CGT GAC CAG GAT ACG ATA TCT rCAT GG/GT3/	Reverse
RH.E8A3CF8496644ACZ0Z.R	T551_02309/GSC-1	129–134	/rhSeq-r/CCT TTA GAA TCT GCA ATA TAT CGT TrGG AAG/GT4/	Reverse
RH.8DF9507B0841432Z0Z.R	mito NAD6	149–153	/rhSeq-r/GCT ATA GCT CCA ACA TAT ACA GTA ATrA TAT G/GT4/	Reverse
RH.3E370038B0CB4B3Z0Z.R	mito NAD1	151–156	/rhSeq-r/GGC AAA ACA AAC CAA ATA GCT TTT TrGT TCT/GT1/	Reverse
RH.DDFCD6AD33494FBZ0Z.R	cytochrome c oxidase subunit 1/ atp8	155–160	/rhSeq-r/TGT CTA GGC AGA ACC CAT TTA rGCG AA/GT2/	Reverse
RH.FF72BC52B1E14BCZ0Z.R	atp6	155–160	/rhSeq-r/ACA AAG AAG GAG AAG AAT TTC AGA rCCA TG/GT1/	Reverse
RH.E67274DA8AD04D9Z0Z.R	cytochrome c oxidase subunit 3	151–156	/rhSeq-r/AGT CCT ACA GAT AAA AAG ATA GCA rCCC AC/GT1/	Reverse
RH.EF9CF32F289649AZ0Z.R	atp9	149–152	/rhSeq-r/GCT TCT GCT AGA GCA AAT CCrC AAG A/GT1/	Reverse
RH.99229D6013E14CBZ0Z.R	NADH dehydrogenase subunit 4L/nad5	151–152	/rhSeq-r/ACA CCG ATT TTA TGT CCT AAC AAA rCCT AC/GT4/	Reverse
RH.795964E1E42344EZ0Z.R	Nad4	125–128	/rhSeq-r/GCC AAA GCC ATA TGT CCA ATA rGAA GA/GT4/	Reverse
i5_IDT-1	N/A	N/A	AAT GAT ACG GCG ACC ACC GAG ATC TAC ACA TAT GCG CAC ACT CTT TCC CTA CAC GAC	rhAmpSeq™ Index 1 (i5) Primer
i5_IDT-2	N/A	N/A	AAT GAT ACG GCG ACC ACC GAG ATC TAC ACT GGT ACA GAC ACT CTT TCC CTA CAC GAC	rhAmpSeq™ Index 2 (i5) Primer
i7_IDT-1	N/A	N/A	CAA GCA GAA GAC GGC ATA CGA GAT ACG ATC AGG TGA CTG GAG TTC AGA CGT GT	rhAmpSeq™ Index 1 (i7) Primer
i7_IDT-2	N/A	N/A	CAA GCA GAA GAC GGC ATA CGA GAT TCG AGA GTG TGA CTG GAG TTC AGA CGT GT	rhAmpSeq™ Index 2 (i7) Primer
i7_IDT-3	N/A	N/A	CAA GCA GAA GAC GGC ATA CGA GAT CTA GCT CAG TGA CTG GAG TTC AGA CGT GT	rhAmpSeq™ Index 3 (i7) Primer
i7_IDT-4	N/A	N/A	CAA GCA GAA GAC GGC ATA CGA GAT ATC GTC TCG TGA CTG GAG TTC AGA CGT GT	rhAmpSeq™ Index 4 (i7) Primer
i7_IDT-5	N/A	N/A	CAA GCA GAA GAC GGC ATA CGA GAT TCG ACA AGG TGA CTG GAG TTC AGA CGT GT	rhAmpSeq™ Index 5 (i7) Primer
i7_IDT-6	N/A	N/A	CAA GCA GAA GAC GGC ATA CGA GAT CCT TGG AAG TGA CTG GAG TTC AGA CGT GT	rhAmpSeq™ Index 6 (i7) Primer
i7_IDT-7	N/A	N/A	CAA GCA GAA GAC GGC ATA CGA GAT ATC ATG CGG TGA CTG GAG TTC AGA CGT GT	rhAmpSeq™ Index 7 (i7) Primer
i7_IDT-8	N/A	N/A	CAA GCA GAA GAC GGC ATA CGA GAT TGT TCC GTG TGA CTG GAG TTC AGA CGT GT	rhAmpSeq™ Index 8 (i7) Primer
i7_IDT-9	N/A	N/A	CAA GCA GAA GAC GGC ATA CGA GAT ATT AGC CGG TGA CTG GAG TTC AGA CGT GT	rhAmpSeq™ Index 9 (i7) Primer
i7_IDT-10	N/A	N/A	CAA GCA GAA GAC GGC ATA CGA GAT CGA TCG ATG TGA CTG GAG TTC AGA CGT GT	rhAmpSeq™ Index 10 (i7) Primer
i7_IDT-11	N/A	N/A	CAA GCA GAA GAC GGC ATA CGA GAT GAT CTT GCG TGA CTG GAG TTC AGA CGT GT	rhAmpSeq™ Index 11 (i7) Primer
i7_IDT-12	N/A	N/A	CAA GCA GAA GAC GGC ATA CGA GAT AGG ATA GCG TGA CTG GAG TTC AGA CGT GT	rhAmpSeq™ Index 12 (i7) Primer

N/A: not applicable.

## Data Availability

Accession to cite these SRA data: PRJNA860648.

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
