# Peer review of "Targeted NGS-Based Analysis of Pneumocystis jirovecii Reveals Novel Genotypes"

_jof, 2022, doi:10.3390/jof8080863_

Round 1

Reviewer 1 Report

The manuscript describes a new targeted NGS-based method for genotyping P. jirovecii clinical isolates. This method involves PCR amplification of multiple nuclear and mitochondrial loci in a total of 36 respiratory samples collected from patients across 6 countries (2-6 samples per countries).  Despite the small sample size, the authors claimed to have identified geographically clustered haplotypes. The manuscript could be improved by addressing the questions and concerns listed below:

Major concerns:

1.     While the cited reference (#5 by the same group for this manuscript) claimed a number of differentially-expressed genes between the ascus and trophic forms based on RNA-seq, only one GSC-1 was further investigated by immunofluorescence assay with antibody against recombinant GSC-1. For all other differentially expressed genes, it remains uncertain if any of them is significantly differentially translated between these two forms as it has been shown in many other organisms that a gene’s mRNA transcription level doesn’t not necessarily always predict its protein level, aside from potential experimental errors/biases in sample preparations and transcriptomic analysis.  In addition, no gene (or genomic DNA loci) has been shown to be unique or specific for any life stage of Pneumocystis while only a very few genes have been experimentally verified to be translated in specific stages.

2.     More details are needed for the methods. Given that rhAmpSeq is a new technology that has not been used for Pneumocystis before, it would be helpful to provide readers with a brief introduction of how it works and what the main procedures and advantages are. It may be necessary to indicate if the DNA polymerase used for PCR has proof-reading activity as regular Taq is error-prone with the potential to introduce artificial SNPs. In the NGS Analysis section, it is necessary to describe how to rule out potential sequence errors introduced by PCR and subsequent NGS process, and what criteria were used to define a SNP (e.g. how many or what percentage of raw reads were used to call a SNP or indel).    

Numerous studies have shown a very high prevalence of coinfection with multiple P. jirovecii strains by genotyping especially NGS (90-100%).  No information on coinfection was given in this manuscript (assuming the SNPs indicated by “>” refer to a nucleotide change between the two nucleotides in front and back of “>” rather than a mixture of two nucleotides). I strongly suspect that there were heterozygous SNPs in some of the targets sequenced, particularly at the positions with SNPs identified. This should be easily revealed by mapping the NGS reads to the reference using SeqMan even under very high stringent conditions (e.g. with >95% matching percentage). The presence of heterozygous SNPs would make it difficult or impossible to determine haplotypes based on SNPs located on different short NGS reads or different genetic loci. The haplotypes shown in the manuscript donot appear to be haplotypes but rather more like genotypes (at individual genetic loci). It would be important to describe how a haplotype was defined in the Methods section.

The description of the Sequencing Analysis section is confusing. It is not clear if any RNA sequencing was performed. If yes, please provide methods for RNA preparation and sequencing; if not, justify the use of RNA-seq/Transcriptomics workflow for DNA-seq data (Line 117).  Given that different DNA targets were amplified by different primers which are expected to have variable amplification efficiencies and that the number of NGS reads is also highly variable between different targets, it is questionable to treat the DNA NGS data as RNA-seq dada and normalize the read count by RPKM (Lines 117-120).

3.     While the authors claimed to have identified geographically clustered haplotypes, the sample size is very small, especially when separated according to geographic origin (up to 6 for 5 out of 6 countries and 11 for one country). With such a small sample size, it appears impossible to do a meaningful statistical analysis. Indeed, no statistical data was provided for any conclusion. Therefore, the geographic association as well as other associations claimed are not valid. All relevant conclusions and discussions need to be revised throughout.     

4.     This manuscript relies primarily on NGS data but no NGS data were made available during review. It is strongly suggested to deposit all relevant NGS raw reads and assembled contigs (if available), along with a metadata file, into a public domain (e.g. NCBI database) before the manuscript is formally accepted for publication.

Minor concerns:

1.     Abstract: Line 16: Give the volume for the gene copy number, and define “copies” (e.g. copies of genome equivalents). Lines 17: Spell NP. Line 19: “life-from specific assays may be useful in non-bronchoscopic diagnosis” reads awkward and needs rewording.

2.     Introduction: As a manuscript focusing on genotyping, it would be helpful to give a brief summary of the currently available typing methods.

3.     Lines 35-36: Inability to culture may be not the primary/sole reason for the underestimated incidence of PCP as I believe that for many pathogens particularly slow-growing pathogens culture-based method should usually be less sensitive than molecular detection methods such as PCR and immunofluorescence assays.

4.     Lines 40-41: Add volume for the copy number and clarify if the detection target is RNA or DNA for the rRNA gene.

5.     Lines 44-45: Spell mtLSU and add volume for the copy number. The term “infection” may be confusing and would be better to be replaced by “disease”. Not clear if the first sentence refers to either or both DNA- and RNA-based assay targeting mtLSU.

6.     Line 55: Define RT-PCR, CRISPR and NGS.

7.     Lines 63-64: Point out the number of samples/patients and if there were sequential samples from the same patients.

8.     Lines 80-83: The phrase “genes enriched in trophs” seems vague and needs to be rephrased as trophs and asci are supposed to share the same genome though with variability in expression level for some genes. Please make it clear that the genes listed for P. jirovecii were based solely on homology with P. murina genes from the cited reference. Also clarify where the mitochondrial genes came from and if they have been confirmed to be preferentially expressed in trophs.

9.     Line 109: Provide the source and accession number of the reference genome used.

10.  Line 120: Spell RPKM.

11.  Lines 135-136: Give possible explanations for the failure of amplification of nad5.

12.  Lines 206-208: Define if “>” indicates a nucleotide change, a mixture of two SNPs or something else.   

13.  Line 209: Not clear what “100%” means exactly.

14.  Lines 213-214: Is this SNP located in the conserved or potentially functional domain of the encoded protein?

15.  Line 227: Replace “position” with “codon”.

16.  Lines 232-233: Couldn’t locate Table IVb.

17.  Line 239: Replace “Table S1 and S2” by “Table S2 and S3”? Results of nuclear genomic SNPs other than those at Meu10 should be summarized in the main text under the Results section.

18.  Line 245: It is recommended to cite the most relevant references for Pneumocystis genomes.

19.  Line 250: Not clear what “single site” refers to exactly.

20.  Lines 244-255: Give volume for the copy number. May be better to replace “(based on major surface glycoprotein PCR)” by “of the major surface glycoprotein gene”?

21.  Line 263: Not clear what “variants” means exactly.

22.  Lines 264-275: Provide the sources/references for all reference genomes used. The description of “S to N variant” and “S to T variant” at meu10 for P. macacae and P. murina is confusing. Does it mean a change between different strains for each of these two species or between P. jirovecii and P. macacae or P. murina? If it’s the latter case, the change may simply reflect a different codon usage preference between different species. These concerns are also applied to the Val to Phe change at atp8 (lines 272-274). Is this SNP located in the conserved/functional domain of the Atp8 enzyme? In addition, given the lack of information on biological and clinical characteristics of the P. jirovecii reference strain, caution should be taken when interpreting the significance of the SNPs identified from clinical samples in this study compared to the reference strain.

23.  Line 296: The statement is not accurate and should be clarified regarding the availability of NGS raw reads.  

Tables:

Table I could be removed as all information appears to be also included in Table S1. Please clarify what the direction of the primer sequence is, what the lowercase r in each primer sequence (near the right end) refers to, how the primers are paired, and what the length range of the amplicon targets. It would be better to use the same style for all gene names/IDs under Target. Explain the question marks under Target.

Table II: All information in this table appears to be included in Tables IV and V as well. If this is true, Table II could be deleted or moved to Supplemental.

Table III: Difficult to read and could be presented in a graph and/or moved to Supplemental.  

Tables IV and V: Both tables could be simplified by using only numbers for ages, one letter for sex (F/M) and one symbol for HIV status (+/-) in the cell and add explanations to the footnote. Explain what “months” refers to and what “>” stands for. Also explain all empty cells and may consider omitting them in the table. Columns 1 and 3-5 in Table IV could be omitted from the table and explained in the title/footnote. Give the gene name for all targets in column 3 of Table V.  Also explain “ins” and “del” in Table V.    

Author Response

Reviewer 1

The manuscript describes a new targeted NGS-based method for genotyping P. jirovecii clinical isolates. This method involves PCR amplification of multiple nuclear and mitochondrial loci in a total of 36 respiratory samples collected from patients across 6 countries (2-6 samples per countries).  Despite the small sample size, the authors claimed to have identified geographically clustered haplotypes. The manuscript could be improved by addressing the questions and concerns listed below:

Major concerns:

  1. While the cited reference (#5 by the same group for this manuscript) claimed a number of differentially-expressed genes between the ascus and trophic forms based on RNA-seq, only one GSC-1 was further investigated by immunofluorescence assay with antibody against recombinant GSC-1. For all other differentially expressed genes, it remains uncertain if any of them is significantly differentially translated between these two forms as it has been shown in many other organisms that a gene’s mRNA transcription level doesn’t not necessarily always predict its protein level, aside from potential experimental errors/biases in sample preparations and transcriptomic analysis. In addition, no gene (or genomic DNA loci) has been shown to be unique or specific for any life stage of Pneumocystis while only a very few genes have been experimentally verified to be translated in specific stages.

We agree with the reviewer that protein validation would be needed for an antigen based diagnostic assay.  However this would not necessarily be required for an RNA based detection assay which is used for several RNA viruses including SAR-CoV2, HIV, and, Hepatitis C.  The goal of this study was to genotype putative RNA targets in Pneumocystis, as a single nucleotide polymorphism could affect the fidelity of a RT-PCR or CRISPR based diagnostic assay.  We have made this clearer in the revised Introduction. 

  1. More details are needed for the methods. Given that rhAmpSeq is a new technology that has not been used for Pneumocystis before, it would be helpful to provide readers with a brief introduction of how it works and what the main procedures and advantages are. It may be necessary to indicate if the DNA polymerase used for PCR has proof-reading activity as regular Taq is error-prone with the potential to introduce artificial SNPs. In the NGS Analysis section, it is necessary to describe how to rule out potential sequence errors introduced by PCR and subsequent NGS process, and what criteria were used to define a SNP (e.g. how many or what percentage of raw reads were used to call a SNP or indel).

This has been addressed in the revised paper.  Below is the additional paragraph that has been added to the Library preparation section of the Methods:

The rhAmpSeq workflow includes two PCR amplifications to generate the libraries. The first PCR includes RNase H-dependent PCR (rhPCR) primers along with RNase H2 enzyme to perform rhPCR where rhPCR, is a nucleic acid amplification method that provides increased target specificity over traditional PCR. As opposed to traditional DNA primers used in PCR and qPCR, rhPrimers contain a single RNA base (rN) and a 3’ blocking group.  This latter moiety can only be cleaved after they anneal to the matched target sequence.  This reduces the risk of non-target amplification such as primer dimers. The amplification was performed using Taq polymerase, as it is not recommended to use a High-Fidelity polymerase since proofreading enzymes with 3’-5’ exonuclease activity can remove the 3’ blocking group which could allow for non-specific amplification. The second PCR in the rhAmpSeq workflow is the indexing PCR which amplifies rhAmp PCR amplicons using i7 and i5 index primers to complete the library building process. The rhAmpSeq system includes purification after both PCR steps using Agencourt Ampure XP Beads (Beckman Coulter, A63880).

Additionally, below is the paragraph added to the NGS Analysis section:

During the workflow for the assembly to generate the SNP table, for the post-assembly analysis options, High SNP filter stringency was used which includes 75% as the minimum variant percentage, and “P not ref” of 99.9%, which is the probability that the base does not match the reference.

Additionally, for all the SNPs reported, the percentage of the sequence at this position in the assembly which varied from the reference (SNP%) ranged from a minimum of 75% to maximum of 100%.      

Numerous studies have shown a very high prevalence of coinfection with multiple P. jirovecii strains by genotyping especially NGS (90-100%).  No information on coinfection was given in this manuscript (assuming the SNPs indicated by “>” refer to a nucleotide change between the two nucleotides in front and back of “>” rather than a mixture of two nucleotides). I strongly suspect that there were heterozygous SNPs in some of the targets sequenced, particularly at the positions with SNPs identified. This should be easily revealed by mapping the NGS reads to the reference using SeqMan even under very high stringent conditions (e.g. with >95% matching percentage). The presence of heterozygous SNPs would make it difficult or impossible to determine haplotypes based on SNPs located on different short NGS reads or different genetic loci. The haplotypes shown in the manuscript donot appear to be haplotypes but rather more like genotypes (at individual genetic loci). It would be important to describe how a haplotype was defined in the Methods section.

We agree with the reviewer and changed haplotypes to genotypes. 

The description of the Sequencing Analysis section is confusing. It is not clear if any RNA sequencing was performed. If yes, please provide methods for RNA preparation and sequencing; if not, justify the use of RNA-seq/Transcriptomics workflow for DNA-seq data (Line 117).  Given that different DNA targets were amplified by different primers which are expected to have variable amplification efficiencies and that the number of NGS reads is also highly variable between different targets, it is questionable to treat the DNA NGS data as RNA-seq dada and normalize the read count by RPKM (Lines 117-120).

This has been addressed. No RNA Sequencing was performed. This description has been removed and replaced with the description used to provide the average coverage depth, and is included below.

The assembly files generated from the previously explained pipeline were used in SeqMan Ultra 17 software which provided the average depth calculation across features. Using the View menu and selecting features, and then the contig of interest, the resulting table provides the average coverage depth for the corresponding genes (Supplemental Table II).

  1. While the authors claimed to have identified geographically clustered haplotypes, the sample size is very small, especially when separated according to geographic origin (up to 6 for 5 out of 6 countries and 11 for one country). With such a small sample size, it appears impossible to do a meaningful statistical analysis. Indeed, no statistical data was provided for any conclusion. Therefore, the geographic association as well as other associations claimed are not valid. All relevant conclusions and discussions need to be revised throughout.

We agree with the reviewer and added to the discussion that this type of analysis would require a larger sample size. 

  1. This manuscript relies primarily on NGS data but no NGS data were made available during review. It is strongly suggested to deposit all relevant NGS raw reads and assembled contigs (if available), along with a metadata file, into a public domain (e.g. NCBI database) before the manuscript is formally accepted for publication.

 This has been addressed. The SRA accession number has been added under Data Availability Statement.

The Accession number for these data in SRA is PRJNA860648

Minor concerns:

  1. Abstract: Line 16: Give the volume for the gene copy number, and define “copies” (e.g. copies of genome equivalents). Lines 17: Spell NP. Line 19: “life-from specific assays may be useful in non-bronchoscopic diagnosis” reads awkward and needs rewording.

This has been addressed and we have reworded this sentence in the revised paper. The threshold that the PERCH investigators used for P. jirovecii infections was defined as 4.0 log10 copies per mL of specimen. PERCH used the Fast Track Diagnostics Respiratory Pathogen 33 (FTD-RP33) panel in their study. We have clarified this in the paper. 

  1. Introduction: As a manuscript focusing on genotyping, it would be helpful to give a brief summary of the currently available typing methods.

We agree and have modified to text to address this.

  1. Lines 35-36: Inability to culture may be not the primary/sole reason for the underestimated incidence of PCP as I believe that for many pathogens particularly slow-growing pathogens culture-based method should usually be less sensitive than molecular detection methods such as PCR and immunofluorescence assays.

We agree and have modified the text to address this.  

  1. Lines 40-41: Add volume for the copy number and clarify if the detection target is RNA or DNA for the rRNA gene.

Detection in the PERCH study was based on PCR using the Fast Track Diagnostics Respiratory Pathogen 33 (FTD-RP33).  This includes a reverse transcription step to detect RNA viruses.  It is likely that in the PERCH study that both DNA and RNA contributed to this signal.  For our study we did not perform reverse transcription so our data derives from DNA. 

  1. Lines 44-45: Spell mtLSU and add volume for the copy number. The term “infection” may be confusing and would be better to be replaced by “disease”. Not clear if the first sentence refers to either or both DNA- and RNA-based assay targeting mtLSU.

We have made these modifications in the revised paper. 

  1. Line 55: Define RT-PCR, CRISPR and NGS.

We have made these modifications in the revised paper.

  1. Lines 63-64: Point out the number of samples/patients and if there were sequential samples from the same patients.

This was a cross sectional study using either PERCH or clinical samples.  We did not to any sequential sample analysis.  We have made this clearer in the revised paper.  

  1. Lines 80-83: The phrase “genes enriched in trophs” seems vague and needs to be rephrased as trophs and asci are supposed to share the same genome though with variability in expression level for some genes. Please make it clear that the genes listed for P. jirovecii were based solely on homology with P. murina genes from the cited reference. Also clarify where the mitochondrial genes came from and if they have been confirmed to be preferentially expressed in trophs.

We thank the reviewer for this comment. We have clarified the text accordingly to note several key points. First, we have changed the phrase above to ‘differentially expressed genes enriched in trophs’ to reflect the reviewer’s point. Our previous identification of these differentially expressed genes came from a RNA sequencing study of separated asci and trophs via flow cytometry. In this manuscript, we validated several targets, including differential expression of SP (enriched in trophs) and Arp9 and GSC-1 (enriched in asci). Second, in regards to the mitochondrial genes, when we performed the enrichment/sequencing above, nearly 50% of the genes differentially expressed in trophs corresponded to mitochondrial genes. This was noted in the 2019 mSphere manuscript. We have not confirmed these mitochondrial targets directly, although that manuscript contains several pieces of data suggestive of this finding. Our primary probe for qRT-PCR measurement of Pneumocystis is the mitochondrial small subunit rRNA. This transcript can still be highly detected in mice treated with micafungin (selectively depleting the asci). Additionally, mice immunized with GSC-1 had reduction in ascus burden by GMS staining and measurement of ascus-enriched transcripts, but continued to have high levels of SSU copy number. These data are supportive, but not confirmative, of the finding that trophs have increased expression of mitochondrial targets. Lastly, we used alignments of P. murina, P carinii, and, P jirovecii to choose our targets for genotyping.

  1. Line 109: Provide the source and accession number of the reference genome used.

This information has been added to the Sequencing analysis section of the Methods, and is included below:

            The reference genome used for the whole genome of Pneumocystis jirovecii is the P. Jirovecii genome assembly strain RU7 (NCBI accession number GCF_001477535.1), reference was downloaded from NCBI as genbank genome file. The reference genome used for the Pneumocystis jirovecii mitochondrial genome is the Pneumocystis jirovecii mitochondrion, complete genome from NCBI; the accession number is NC_020331 and the version number is NC_020331.1. This genome was downloaded from NCBI in GenBank format.

  1. Line 120: Spell RPKM.

We have removed RPKM. 

  1. Lines 135-136: Give possible explanations for the failure of amplification of nad5.

The target that the primers of nad5 & nad4l are targeting is the intergenic regions which did amplify. This is similar to how cox1 & atp8 primers targeted and amplified the intergenic regions between the two genes. We have corrected this in the SNP data and have clarified this in the text. 

  1. Lines 206-208: Define if “>” indicates a nucleotide change, a mixture of two SNPs or something else.

When a SNP includes “>”, it indicates the directionality of the nucleotide change, going from reference sequence to variant sequence. “C>T” indicates T was the nucleotide in the reference genome and C is the nucleotide change of the variant sequence which defines the SNP.

This explanation was added as “SNP Legend” below all the SNP tables including Table II, Table III, Supplemental Table 3, and Supplemental Table 4.

  1. Line 209: Not clear what “100%” means exactly.

This refers to the concordance of the two SNPs in Meu10.  There were no instances where only one SNP was present and the other absent. 

  1. Lines 213-214: Is this SNP located in the conserved or potentially functional domain of the encoded protein?

By definition the targets chosen for genotyping were driven by conservation of the targets between P. murina, P carinii, and, P jirovecii.  For SNPs identified in the coding sequence we conducted potential functional analyses with the PredictSNP tool   (https://loschmidt.chemi.muni.cz/predictsnp/).   This analyses showed that the likelihood of a deleterious effect was unlikely.    

  1. Line 227: Replace “position” with “codon”.

This has been addressed.

  1. Lines 232-233: Couldn’t locate Table IVb.

This has been addressed.

  1. Line 239: Replace “Table S1 and S2” by “Table S2 and S3”? Results of nuclear genomic SNPs other than those at Meu10 should be summarized in the main text under the Results section.

This has been addressed. Results of nuclear genomic SNPs other than those at Meu10 have been added for Four F5.  The other SNPs (due to their lower frequency are summarized in Supplementary Tables III and IV.  

  1. Line 245: It is recommended to cite the most relevant references for Pneumocystis genomes.

We have added additional references to address this.  

  1. Line 250: Not clear what “single site” refers to exactly.

This refers to a single clinical site/hospital.  We have clarified that in the revised paper. 

  1. Lines 244-255: Give volume for the copy number. May be better to replace “(based on major surface glycoprotein PCR)” by “of the major surface glycoprotein gene”?

We have modified the text to address this concern. 

  1. Line 263: Not clear what “variants” means exactly.

Variants is used synonymously with SNPs.

  1. Lines 264-275: Provide the sources/references for all reference genomes used. The description of “S to N variant” and “S to T variant” at meu10 for P. macacae and P. murina is confusing. Does it mean a change between different strains for each of these two species or between P. jirovecii and P. macacae or P. murina? If it’s the latter case, the change may simply reflect a different codon usage preference between different species. These concerns are also applied to the Val to Phe change at atp8 (lines 272-274). Is this SNP located in the conserved/functional domain of the Atp8 enzyme? In addition, given the lack of information on biological and clinical characteristics of the P. jirovecii reference strain, caution should be taken when interpreting the significance of the SNPs identified from clinical samples in this study compared to the reference strain.

These are amino acid changes predicted from the DNA sequence.  We have made this clearer in the revised paper that these are predictions. 

  1. Line 296: The statement is not accurate and should be clarified regarding the availability of NGS raw reads.

This has been addressed. The SRA accession number has been added to the data availability statement.

Tables:

Table I could be removed as all information appears to be also included in Table S1. Please clarify what the direction of the primer sequence is, what the lowercase r in each primer sequence (near the right end) refers to, how the primers are paired, and what the length range of the amplicon targets. It would be better to use the same style for all gene names/IDs under Target. Explain the question marks under Target.

This has been addressed. Table I has been revised to include the target length. The primers are paired in that each target has both a Forward and a Reverse primer, which are noted in the Primer Notes on the right. There are two pairs of targets, cox1 & atp8 and nad4l &nad5, and each pair share forward and reverse primers, in that the primers amplify the intergenic regions between the two targets. Additionally, under Gene name, the gene name was added to the primers which were previously only labeled by their gene ID. The question marks have been removed; those were not meant to be included in the table. We have included all the information necessary in the revised Table I and will remove Supplemental Table 1, as it is no longer needed. Lastly, the lowercase r was explained in the Methods section.

Table II: All information in this table appears to be included in Tables IV and V as well. If this is true, Table II could be deleted or moved to Supplemental.

 Table II (demographic information of the samples) has been moved to the Supplemental section and is now Supplemental Table 1.

Table III: Difficult to read and could be presented in a graph and/or moved to Supplemental. 

We think Table III is presented best entirely and has been moved to the Supplemental data section and is now Supplemental Table 2.  We can provide a PDF or XLS file for readers to download to facilitate independent review. 

Tables IV and V: Both tables could be simplified by using only numbers for ages, one letter for sex (F/M) and one symbol for HIV status (+/-) in the cell and add explanations to the footnote. Explain what “months” refers to and what “>” stands for. Also explain all empty cells and may consider omitting them in the table. Columns 1 and 3-5 in Table IV could be omitted from the table and explained in the title/footnote. Give the gene name for all targets in column 3 of Table V.  Also explain “ins” and “del” in Table V. 

This has been addressed. The tables have been simplified and each have a demographics legend below the tables. “Months” refers to the ages of the samples under 1 year of age, and therefore expressed by age in months. A SNP legend has been added below the tables explaining “>” indicates the directionality of the nucleotide change of the SNP. “ins” and “del”, which indicate an insertion of nucleotide or deletion of nucleotide, respectively; however, the rows which had insertion or deletion SNPs were removed from Table V, since they were not necessary for the discussion of the mitochondrial SNPs. The repetitive columns were deleted and summarized in the title of each table. For the gene name of Table V, the gene name for a row is blank as that position is in the intergenic regions between cox1 and atp8, which is now explained in the title of the table.

Table IV is now Table II.

Table V is now Table III.

Reviewer 2 Report

Dear authors,

It has been a pleasure to review the manuscript “Targeted NGS-based Analyzes of Pneumocystis jiroveci reveals geographically clustered haplotypes” by Dora Pungan and colleagues, submitted for publication in Journal of Fungi. This article is an important contribution to the development of P. jiroveci detection tools and to the knowledge of the epidemiology of Pneumocystosis. In spite of the accurate quality of this important work (the research), the document (the manuscript) that describes it needs to be improved. In that sense, I have some comments and suggestions for the authors. I am going to mention those comments and suggestions in the same order they appear in the manuscript:

 - In the introduction, between line 32 and 33, the authors affirm “Pneumocystis jiroveci is the leading cause of pneumonia in young children regardless of HIV status”. An important cause…, yes; but the leading cause …, not. I suggest the authors to revise the article they cite (Pneumonia Etiology Research for Child Health Study, G., Causes of severe pneumonia requiring hospital admission in 303 children without HIV infection from Africa and Asia: the PERCH multi-country case-control study. Lancet 2019, 394, (10200), 304 757-779).

 - In the introduction, between line 35 and 37, the authors affirm “Due to the inability to culture Pneumocystis sp., the epidemiology of this infection likely underestimates the true incidence and prevalence of this infection”. This is not incorrect, but I suggest “Due to the inability of available diagnostic procedures…” as it has not been only because of the inability to culture Pneumocystis sp.

 - The abbreviation mtLSU appears by the first time in the introduction (line 44). Nevertheless, the description of that abbreviation appears by the first time in the discussion, “…the mitochondrial large subunit of ribosomal RNA (mtLSU)”, between line 35 y 37. It must be corrected.

 - In the introduction, between line 45 and 50, the authors must mention, and reference, to the potential reader, that ascus is more frequent in colonization and the troph more frequent in infection.

 - The authors should refer to the limitations of the study; for example, number of samples, and representativeness of the number of countries in the samples.

 - In the discussion, I suggest to the authors refer to the potential limitations that mitochondrial heteroplasmy could bring to the work with mitochondrial genes.

 - The end of the manuscript is some abrupt. Aspects addressed in the introduction are not addressed at the exit.

Author Response

Reviewer 2

It has been a pleasure to review the manuscript “Targeted NGS-based Analyzes of Pneumocystis jiroveci reveals geographically clustered haplotypes” by Dora Pungan and colleagues, submitted for publication in Journal of Fungi. This article is an important contribution to the development of P. jiroveci detection tools and to the knowledge of the epidemiology of Pneumocystis. In spite of the accurate quality of this important work (the research), the document (the manuscript) that describes it needs to be improved. In that sense, I have some comments and suggestions for the authors. I am going to mention those comments and suggestions in the same order they appear in the manuscript:

 - In the introduction, between line 32 and 33, the authors affirm “Pneumocystis jiroveci is the leading cause of pneumonia in young children regardless of HIV status”. An important cause…, yes; but the leading cause …, not. I suggest the authors to revise the article they cite (Pneumonia Etiology Research for Child Health Study, G., Causes of severe pneumonia requiring hospital admission in 303 children without HIV infection from Africa and Asia: the PERCH multi-country case-control study. Lancet 2019, 394, (10200), 304 757-779).

We thank the reviewer for this comment.  We accidently omitted the world fungal prior to pneumonia. 

 - In the introduction, between line 35 and 37, the authors affirm “Due to the inability to culture Pneumocystis sp., the epidemiology of this infection likely underestimates the true incidence and prevalence of this infection”. This is not incorrect, but I suggest “Due to the inability of available diagnostic procedures…” as it has not been only because of the inability to culture Pneumocystis sp.

We have modified this in the revised paper to highlight the multiple factors that lead to under diagnosis.

 - The abbreviation mtLSU appears by the first time in the introduction (line 44). Nevertheless, the description of that abbreviation appears by the first time in the discussion, “…the mitochondrial large subunit of ribosomal RNA (mtLSU)”, between line 35 y 37. It must be corrected.

This has been addressed in the revised paper.

 - In the introduction, between line 45 and 50, the authors must mention, and reference, to the potential reader, that ascus is more frequent in colonization and the troph more frequent in infection.

This has been addressed.

 - The authors should refer to the limitations of the study; for example, number of samples, and representativeness of the number of countries in the samples.

 - In the discussion, I suggest to the authors refer to the potential limitations that mitochondrial heteroplasmy could bring to the work with mitochondrial genes.

 - The end of the manuscript is some abrupt. Aspects addressed in the introduction are not addressed at the exit.

This is an excellent point and we have addressed this in the revised discussion. 

One of our author’s name (Simran K Arora), had the wrong middle initial and prefix in our original submission. The correct name is Simran K Arora (this has been fixed in the revised paper); and her correct prefix is Dr.